# The Brazilian System for Monitoring Workers and General Population Exposed to Asbestos: Development, Challenges, and Opportunities for Workers’ Health Surveillance

**DOI:** 10.3390/ijerph20054295

**Published:** 2023-02-28

**Authors:** Rafael Junqueira Buralli, Regina Dal Castel Pinheiro, Laura Lima Susviela, Sandra Renata Canale Duracenko, Eduardo Mello De Capitani, Alexandre Savaris, Eduardo Algranti

**Affiliations:** 1Departamento de Vigilância em Saúde Ambiental e Saúde do Trabalhador, Ministério da Saúde, Brasil (CGSAT/DSAST/SVS/MS), SRTV 702, Via W5 Norte, Brasília 70723-040, DF, Brazil; 2Gerência em Saúde do Trabalhador, Diretoria de Vigilância Sanitária do Estado de Santa Catarina (GESAT/DIVS/SUV/SES), Av. Rio Branco, 152, Centro, Florianópolis 88015-200, SC, Brazil; 3Centro de Referência em Saúde do Trabalhador de Piracicaba (Cerest Piracicaba/SMS/PMP), Rua do Trabalho, 634, 1º andar, Vila Independência, Piracicaba 13418-220, SP, Brazil; 4Departamento de Clínica Médica, Centro de Informação e Assistência Toxicológica (CIATox) de Campinas, FCM, Universidade Estadual de Campinas, Rua Vital Brasil 251, Campinas 13083-888, SP, Brazil; 5Fundação de Amparo à Pesquisa e Extensão Universitária (FAPEU), Rua Delfino Conti, S/N, Florianópolis 88040-370, SC, Brazil; 6Division of Applied Research, Fundacentro, Rua Capote Valente, 710, São Paulo 05409-002, SP, Brazil

**Keywords:** asbestos, occupational health, public health surveillance, lung diseases, asbestosis, mesothelioma

## Abstract

The lack of safe levels of asbestos exposure and the long latency of asbestos-related disease (ARD) makes workers’ health surveillance challenging, especially in lower-income countries. This paper aims to present the recently developed Brazilian system for monitoring workers and general population exposed to asbestos (Datamianto), and to discuss the main challenges and opportunities for workers’ health surveillance. Methods: a descriptive study of the Datamianto development process, examining all the stages of system planning, development, improvement, validation, availability, and training of health services for its use, in addition to presenting the main challenges and opportunities for its implementation. Results: The system was developed by a group of software developers, workers’ health specialists, and practitioners, and it was recently incorporated by the Ministry of Health to be used for workers’ health surveillance. It can facilitate the monitoring of exposed individuals, epidemiological data analysis, promote cooperation between health services, and ensure periodical medical screening guaranteed to workers by labor legislation. Moreover, the system has a Business Intelligence (BI) platform to analyze epidemiologic data and produce near real-time reports. Conclusions: Datamianto can support and qualify the healthcare and surveillance of asbestos-exposed workers and ARD, promoting a better quality of life for workers and improving companies’ compliance with legislation. Even so, the system’s significance, applicability, and longevity will depend on the efforts aimed at its implementation and improvement.

## 1. Introduction

Occupational asbestos exposure is a significant public health problem globally, resulting in more than 233,000 deaths per year and inestimable socioeconomic impacts [1]. Various diseases are associated with asbestos exposure, such as asbestosis, pneumoconiosis, lung cancer, mesothelioma, and other cancers [2,3,4,5,6]. In Brazil, workers from the asbestos-cement industry have a higher mortality ratio due to all cancer types, and also pleural, peritoneal, and lung cancer, as well as asbestosis [7].

Brazil is historically one of the world’s largest producers, exporters, and consumers of asbestos, primarily chrysotile asbestos employed in the manufacture of asbestos-cement products for construction purposes, which are mainly used to produce roof tiles and water tanks. Brazil’s asbestos is mostly exported to Asia and Latin America, or used to supply the declining domestic market, leading to millions of previously and currently exposed workers [8]. Considering the asbestos consumption pattern in Brazil and asbestos-related disease’s (ARD) long-latency, which can be 30 years or more, it was estimated that mesothelioma cases would peak between 2021 and 2026 [9].

In Brazil, workplace control measures such as the establishment of tolerance limits, criteria for removal and demolition of materials containing asbestos, the prohibition of children aged under 18 years from working when exposed to asbestos, and use of amphiboles, were not implemented in asbestos-related industries until the 1980s, decades after high levels of asbestos consumption [7]. Later, Brazil banned asbestos mining, processing, commercialization, and distribution, starting with amphibole asbestos in 1991 [10], which was ratified by the Federal Law nº 9.055/1995 [11]. The ban was expanded to chrysotile in 2017 following a legal dispute in the Federal Supreme Court, decided by the Unconstitutionality Direct Actions nº 3406 and 3470 [12].

Brazilian Labor Regulatory Standards nº 15 (NR-15), which regulates chemical threshold limit values, including asbestos, establishes that companies must provide regular medical screening for all exposed workers. The screenings should happen periodically as well as at the time of hiring and dismissal and should include clinical assessment, pulmonary function test (spirometry), and chest X-rays (CXR) interpreted according to the International Labor Organization (ILO) standard, while also informing workers of their results. Companies must maintain this health screening for 30 years after the end of the worker’s contract, as outlined by legislation aiming to cover the ARD latency period [13,14]. Moreover, the Brazilian Law nº 9.055/1995 determines that companies that use or used asbestos must share information annually regarding all exposed workers, such as job held; and dates of birth, admission, and periodic medical evaluation; as well as any clinical diagnoses, with the health surveillance services of the Unified Health System (SUS), as well as with the workers’ representative unions [11].

The long latency of ARD and the lack of safe exposure levels hampers the longitudinal follow-up of exposed workers, the clinical association between exposures and outcomes, and the adoption of surveillance strategies focusing on the exposed population. There are several other challenges for conducting worker health surveillance in Brazil, such as a lack of human and financial resources and well-defined reference health services for occupational diseases. Many workers experienced short-term exposures (e.g., a few months or years), which are difficult to track. In addition, many companies went bankrupt, stopped tracking exposed workers or conducting clinical examinations, or do not share the results with health services. Moreover, the asbestos regulations still need to be fully implemented in Brazil and expanded across the country.

In this context, a group of software developers, workers’ health specialists, and practitioners designed a monitoring system for individuals exposed to asbestos (Datamianto; “Sistema Brasileiro de Monitoramento de Trabalhadores e Populações Expostas ao Amianto” in Portuguese). The system can be used for recording data on health assistance and surveillance and can facilitate the monitoring of exposed workers by the Occupational Health Reference Centers (Cerest), which are multi-professional centers specializing in workers’ health, and which are responsible for implementing the National Policy of Workers’ Health. Datamianto opens the possibility of adding information for the proper follow-up of exposed workers, facilitating an active surveillance. Furthermore, it has a Business Intelligence (BI) platform for analyzing epidemiological data, and visualizing the information in alert panels, and can produce reports to support decision-making. The system also promotes information exchange among health services and helps ensure follow-up appointments and examinations guaranteed to workers by labor legislation.

Epidemiological surveillance systems are crucial for monitoring and controlling ARD; however, currently they are available in only a few countries such as Italy, Australia, France, and South Korea. These systems are crucial to support health services as they can help to track epidemiological situations, identify sources of asbestos contamination and exposure, plan and conduct preventive measures, make diagnoses, and contribute to compensating work-related diseases [15]. However, to succeed they must have reliable information, comprehensive exposure assessment, and consistent territorial coverage [15].

Thus, this article aims to describe the development of a monitoring system for workers and the general population exposed to asbestos in Brazil and to discuss the main challenges and opportunities for worker health surveillance.

## 2. Materials and Methods

This descriptive article presents Datamianto’s development process and operating logic for worker health surveillance in the Brazilian public health system (SUS). All stages of system development are detailed, including:

(a) System planning and fundraising: this describes how Datamianto was planned, the institutions involved in its proposal and funding, and how resources were designated to enable the system’s development.

(b) System development, validation, and improvement: this describes how Datamianto was developed by the development group (DG). User profile creation is presented, as well as the process for validating and improving the system. This section also presents a feature that allows asbestos-related companies to include periodic workers’ health assessments data.

(c) System availability and training of health services’ staff: this presents the system’s development versions, and its later transference to the Ministry of Health datacenter. It also describes how potential Datamianto users were trained by the DG, including Cerest representatives, surveillance and health care centers, universities, and decision-makers. The DG was also in charge of the user support and training process, which included several online meetings, calls, emails, telephone messages, and tutorials.

Furthermore, an overview of workers’ health surveillance in Brazil is provided, including the actors involved and their responsibilities, as well as how Datamianto is expected to be used for monitoring exposed workers considering the SUS structure and capabilities. To illustrate the system’s functionalities and usability by health services, screenshots of the system’s interfaces and BI are provided, including health appointment alerts in line with the regulatory norms, different tabs for surveillance, and health care access and data inclusion. Figures were translated to English for illustration. Furthermore, an overview of preliminary non-identified data entered onto the system by the Cerest of the State of Santa Catarina (Cerest/SC) is shared as an example. Additionally, the main challenges and opportunities offered by the system for the implementation of comprehensive health care for workers exposed to asbestos in Brazil are presented.

## 3. Results and Discussion

### 3.1. Planning and Fundraising for Datamianto Development

In 2019, the Brazilian Association of Poison Control Centers and Clinical Toxicologists (ABRACIT) and the Public Ministry of Labor (MPT) joined forces to develop the Datamianto system. The MPT allocated financial resources for developing and validating a data management tool to be used as a national database for recording treatment, follow-up, and monitoring of individuals exposed to asbestos. The Datamianto system was nested inside the Datatox system, which is used by the national net of Poison Control Centers to record poisoning cases. A workers’ health specialist outlined the initial changes to the Datatox system needed for assessing an individual exposed to asbestos: work that was later expanded on by a group of specialists.

### 3.2. System Development, Validation, and Improvement

From 2019–2022, Datamianto was developed by a DG formed by software developers, information technologists, workers’ health specialists, health care and surveillance practitioners, pulmonologists, stakeholders, and other specialists, with technical support from the General Coordination of Occupational Health Surveillance (CGSAT) of the Brazilian Ministry of Health.

Through weekly or biweekly meetings, the DG and external advisors discussed the system’s features, usability, and how to improve it, considering its application for health services. The system’s updates were presented and discussed, and new features were successively validated, developed, and included in new releases. Considering the Datatox system as a baseline, in total, 173 structural changes were made to its database, 173 backend and/or frontend changes were made to the BI, and 382 changes were made to the system’s interfaces, aiming to meet Datamianto requirements.

The system’s updates were presented and discussed, and new features were successively validated and improved. In total, 173 changes were made to the system’s database, 173 to the BI, and 382 to the system’s interfaces. Two mirrored versions of the system were deployed simultaneously and were hosted initially at the Federal University of Santa Catarina (UFSC), one with real data from exposed workers (pilot version) and another for review, testing and development, as well as for training of users before using the pilot version.

The system was built from the perspective of the SUS and the Brazilian Network for Workers’ Health Surveillance (Renast): a network created in 2002 to organize workers’ healthcare, surveillance, and preventive actions in Brazil, promoting the integration of the Cerest system with other SUS health services [16]. Health surveillance and healthcare are complementary and structuring axes of the SUS policies and services, reinforced by the National Policy on Workers’ Health [17]. Furthermore, the recently published “National action plan for structuring the network of health actions and services for an integrated health care for populations exposed to asbestos” proposes the organization of strategies related to asbestos exposure considering these two axes [18].

There are currently 212 Cerest centers specializing in workers’ health in Brazil, covering all states and hierarchically classified as state, regional, or municipal. This classification was used to determine the user profiles of access to the information in the system. For instance, state-level Cerest and health service professionals can access only information on workers from their state; likewise, access for regional and municipal Cerest and health services professionals is limited to their respective jurisdictions. If needed, patient records can be shared between health services upon request.

Datamianto has two modules related to the type of health service that will be or that was provided to the patient, namely:Assistance: specialized health services related to the diagnosis of ARD, clinical follow-up, and treatment realized by primary, secondary, and tertiary care, which includes conducting periodic CXR, interpreted according to ILO Classification of Pneumoconiosis Radiographs, and spirometry.Surveillance: health promotion and disease prevention activities, considering workers’ health and safety guidelines, NR-15 regulations, and health surveillance related to companies’ obligations.

The system can produce different visual alerts containing the number of cases in each situation in both modules (assistance and surveillance). In the assistance module, users can view the number of patients with a definite diagnosis of ARD, cases under investigation, and cases of death not yet defined as being related to asbestos exposure (Figure 1). A single click on an individual alert exhibits a list of the related cases, which in turn can be accessed individually for maintenance or evaluation.

Brazilian regulations (NR-7 and NR-15) set requirements for companies that used asbestos in their production process, define threshold limits for asbestos exposure, and detail mandatory surveillance actions for companies with exposed workers. Companies using asbestos must conduct an environmental assessment of asbestos dust in the workplace at least every semester. All exposed workers must undergo medical examinations upon hiring, dismissal, and annually, through clinical evaluations, CXR, and spirometry. Employers must guarantee information and training to workers at least annually, prioritizing risk protection and control measures. Employers must perform this assessment of exposed workers for 30 years following termination of employment contracts, at a frequency of either (a) every three years for workers exposed from 0–12 years; (b) every two years for workers exposed from 12–20 years; or (c) annual for workers exposed for more than 20 years. Mandatory data related to workers’ health must be shared with SUS technical references. Periodic medical examinations of exposed workers must be appropriately conducted by the companies or provided by public health services [13,14].

In the surveillance module, it is possible to select a specific company to monitor their obligations to workers occupationally exposed to asbestos, according to the regulations. Moreover, initial alerts show the number of workers who must undergo health assessment every one, two, or three years according to their exposure duration, as well as the number of workers with delayed evaluations, with the end of exposure over 30 years ago, and deaths (Figure 2).

The system allows for the possibility of sharing clinical data between diagnostic centers and specialized health services. This not only promotes the exchange of information among health services, but also helps avoid case misclassification and duplicate examinations and evaluations, such as imaging studies, which may entail additional radiation exposures and additional costs for companies and public health services.

Workers’ data are regularly assessed by the health services professionals, such as those from Cerest and other technical references for worker health surveillance. Before entering new information in the system, it is possible to search existing registries through the patient’s National Health Card (CNS), which contains SUS user data. In this way, duplications are avoided, and the patient’s basic information is automatically filled in, such as the date of birth, address, mother’s name, and personal identification number.

Datamianto users can access information through its 13 tabs: patient’s personal information; work history; exposure information; agent (type of asbestos); symptoms and reevaluation tabs for health practitioners from the assistance services to record patient clinical information; complementary examinations; attachments, where it is possible to attach examination results and relevant documents in different formats, such as pdf, jpeg, mp4, and dcm for radiological images; patient’s diagnosis; CAT (includes the registry number of the work accident report) and Sinan (notifiable diseases information); and information for closing the patient record. Moreover, there is a health surveillance tab used exclusively by Cerest and/or other surveillance services (Figure 3). An English translated version of the patient’s assessment form is provided as Appendix A.

For instance, opening the worker’s form in the system (assistance module) and filling in basic information on history, clinical manifestations, test results (chest X-ray and spirometry), and diagnosis takes around 20 min, which can be increased depending on the length of the patient’s history and the number of complementary examinations to be entered, for example.

The system also allows companies that use or have used asbestos to include data on periodic workers’ health assessments. In this case, the system’s version is more restrictive regarding functions and data visualization. According to the legislation, these companies must conduct clinical reassessment and periodic examinations of their workers. It enables the insertion of follow-up information by a company’s designated technical manager registered in the system. The State of Santa Catarina has used a similar strategy with its information systems, proving to be very promising in facilitating the workers’ surveillance process. It allows the company to monitor workers, while the SUS services are responsible for checking inserted data and performing in loco inspections. This feature has already been developed for Datamianto, but it is still under evaluation and is not currently in use, as the system’s primary focus is supporting the SUS services.

The development of Datamianto was strongly focused on the health surveillance of workers and former workers exposed to asbestos. Still, it can be used for health surveillance of the general population non-occupationally exposed to asbestos and other dust contaminants, such as those living near a source of asbestos pollution or cohabitating with an exposed worker.

Furthermore, a critical tool developed for data management and visualization is the system’s BI module, which can extract data and generate outputs such as tabular reports, dynamic tables, charts, and georeferenced maps customized according to the selected variables. It will enable users to have specific views of the data according to their needs, as can be seen in Figure 4.

For comparison, two similar surveillance systems are the Italian National Register of Malignant Mesotheliomas (ReNaM), and the French National Mesothelioma Surveillance Program (NMSP), although they are more focused on mesothelioma surveillance [15,19]. ReNaM enables a permanent screening of mesothelioma cases and circumstances related to asbestos exposure by the Regional Operating Centres (CORs). The CORs search for incident cases from healthcare institutions such as chest surgery wards, pathology, and lung care units, in order to apply structured questionnaires to patients or family members on environmental and occupational history and lifestyle habits. They also exchange information with health and safety agencies, and with the Social Security Institute (INPS) [15].

NMSP is a French surveillance system established in 1998 to help investigate mesothelioma trends and the proportion attributable to occupational exposure to asbestos. The system also serves to improve pathology diagnosis, its classification as a compensable occupational disease, and to contribute to epidemiological research. Each registered mesothelioma case undergoes an exposure assessment, and a pathological and clinical diagnostic procedure when classified as inconclusive. It also encompasses a case-control survey to explore associated risk factors and helps with the right to compensation from occupational diseases, if applicable [19].

Taken together, there are some similarities between the proposed Brazilian system and the established ReNan and NMSP systems, as they can play a central role in qualifying the active surveillance for asbestos exposure. These systems also help estimate the epidemiological situation, identifying possible sources of contamination and preventing asbestos exposure. Moreover, this comparison points to the need to promote cooperation between health and social security authorities in Brazil, and to the role Datamianto could play in supporting the recognition of ARD as compensable occupational diseases.

### 3.3. Availability, and Training of Health Services

Datamianto was developed to be used by the Ministry of Health to support the implementation of the National Asbestos Plan [18]. To this end, the information inserted onto the system must be used to support the workers’ health analysis, including the recognition of local/regional epidemiological and productive profiles related to asbestos exposure and ARDs, together with other health and work information systems and databases used for surveillance and assistance actions [18].

After stability tests, the system’s development version hosted at UFSC was transferred to an infrastructure provided by the Ministry of Health to host the official version (https://datatox-amianto.aids.gov.br/datatoxamianto/login, accessed on 22 December 2022). At the same time, the official internalization of this system is being processed by DataSUS, the body responsible for all SUS information systems and which collects, processes, and disseminates public health information in Brazil.

For security reasons, a temporary external backup of the database used by Datamianto will continue to be performed until March 2023 to avoid information loss. This action was undertaken due to instabilities and access problems reported by users. Every Datamianto user must previously fill out and sign terms of responsibility and confidentiality in compliance with the General Data Protection Law (LGPD) in effect in Brazil. These terms also must be signed by the health service managers. Handling information about companies and workers’ health and exposure is part of the duties of health professionals from surveillance and healthcare services.

Surveillance and healthcare workers were trained by members of the DG through at least two virtual meetings to present the system’s management, administration, alerts, and functionalities and to discuss its use. Hence, these meetings became a space for discussing and building the best strategies to promote comprehensive health care for workers. In total, 321 participants from 15 of the 27 Brazilian States were trained, mainly workers from Cerest and specialized health services from priority states (those with recognized companies that use or have used asbestos). Moreover, support by email and telephone was provided, and a system’s instruction manual and a tutorial for worker health surveillance professionals were made available.

Datamianto’s use by health services is increasing in Brazil. As of December 2022, it had been used by 27 health services to register information on 20,368 individuals exposed to asbestos, including 10 state Cerest or worker health coordination, 12 regional or local Cerest, and 5 specialized health services, from 9 out of 27 states in Brazil. Moreover, 4882 workers’ surveillance actions were recorded, mainly related to the insertion and verification of whether the company complied with legislation. Health services also registered information on 4405 ARD notifications, with 3043 cases under investigation and 485 confirmed. Moreover, 853 deaths of workers previously exposed to asbestos were recorded, most with the causal relation still under investigation.

Currently, there is no implementation plan for the system, although its use is foreseen in the National Asbestos Plan [18]. However, the Brazilian Ministry of Health is promoting Datamianto’s use to the SUS health services, especially to the workers’ health surveillance network, which is liaising with state and local specialized health services to adopt the system and engage health workers for its use [18]. SUS governance levels (federal, state and municipal) are independent and should work collaboratively and liaise. The Ministry of Health gives guidelines, facilitates, and co-finances programs’ execution, while states and municipalities are responsible for co-financing, coordinating, and implementing actions at local level [16]. Cerest and specialized health services, such as tertiary services, pulmonology, and cancer centers, are responsible for executing health assistance and surveillance in the SUS. Thus, trained health workers from these reference services introduce patient data onto the system.

Considering the high use of asbestos in Brazil, ARD’s long-latency, and the high burden for workers even after short-term exposures, there is an urgent need to identify exposed populations and monitor their health status continuously, providing comprehensive healthcare and surveillance actions [18]. A national ecological analysis comparing mortality in Brazilian municipalities that housed asbestos mining and/or asbestos-cement industries with other national municipalities showed an excess of deaths from mesothelioma and lung cancer in men and from lung and ovarian cancer among women [20].

### 3.4. Example of Datamianto’s Use for the Workers’ Health Surveillance by the SUS

Monitoring the health of workers exposed to asbestos is the responsibility of the company that exposed the worker to the risk of handling or coming into contact with asbestos in work environments and processes. Yearly, companies must send a list of their employees to SUS and also to the workers’ respective unions, indicating each worker’s sector, function, position, date of birth, admission, and periodical medical evaluation and results, in accordance with Brazilian Law nº 9055/1995 [11]. These data are important for the longitudinal monitoring of exposed workers’ health.

Since 2016, Cerest/SC has been following the mandatory actions carried out by asbestos companies in the State of Santa Catarina. More recently, the Datamianto surveillance module is being used for workers’ monitoring, inserting information on companies, workers, asbestos exposure, job held, and health-related data. The system is also used to monitor the CAT (notification of occupational accidents) and Sinan (SUS information system for disease and accidents notification) records by inserting data related to the workers exposed to asbestos.

Currently, Cerest/SC’s work is focused on the regular monitoring of workers exposed in the asbestos-cement industries. Until December 2022, 3528 workers and former workers of a mapped asbestos-related company were included on the system. It is possible that more exposed workers will appear in the state, from this company or from others, as a result of active surveillance, but the total number cannot be estimated. At least 4581 health monitoring actions have already been carried out for workers exposed to asbestos, with 90.9% being carried out by the companies and 9.1% being carried out by Cerest/SC. Moreover, 413 death records were registered on the system by cross-referencing the workers database with the Brazilian Mortality Information System (SIM). Most deaths (76.0%) have not yet been investigated, and of those investigated, 12.1% were not asbestos-related, 7.3% were due to asbestos exposure, and 4.6% are still under investigation. The records also include 11 deaths from COVID-19 and 13 cases with other ARD, including 6 cases of lung cancer, 4 cases of larynx cancer, and one each of asbestosis, non-asbestos interstitial disease, and pleural mesothelioma.

Data on exposed workers also disclosed 1693 workers (48.0%) with an end of exposure date greater than 30 years; therefore, these workers are out of the industry periodical check and must continue to be monitored by the SUS health services. In addition, the data show there are 1422 exposed workers requiring periodic evaluation by companies in the state of Santa Catarina, in compliance with NR-15, and who require follow-up by Cerest/SC, mainly the follow-up of examinations and medical appointments in the SUS. The breakdown of these workers is as follows:131 workers (9.2%) to be assessed annually (exposed >20 years).182 workers (12.8%) to be assessed every 2 years (exposed 12–20 years).1109 workers (78.0%) to be assessed every 3 years (exposed <12 years).

Furthermore, it was identified that companies were late in conducting the reassessments of workers that should be carried out annually (95.4%), every two years (69.2%), and every three years (75.9%). This overview brings into stark contrast instances of non-compliance with legislation regarding the periodicity of health assessments by companies and their adherence to correct information sharing practices.

### 3.5. Main Challenges and Opportunities for Workers’ Health Surveillance

There are several organizational and operational challenges related to the system’s implementation, use, and refinement, as well as for managing the healthcare and surveillance of asbestos exposed workers in Brazil.

Firstly, it is crucial for the Ministry of Health to define how the system should be managed and used, to promote its use, and to make it available for the health services involved in patient care at different levels. It is necessary to ensure the system’s regular maintenance and update, and to provide sufficient human resources and technological capacity for health services. It is also important to provide adequate training and support to the health services for the planning and organizing of healthcare and surveillance actions, evaluation of periodic examinations, and referrals to tertiary care. In this sense, the Ministry of Health needs to propose adequate procedures for information flow, data collection, and analysis, as well as a health communication plan, which must be supervised by a technical group formed by the Ministry of Health.

An information registry system may support epidemiological research on asbestos exposure pathways and ARD etiology, including the development of case-control and cohort studies, prioritizing current and former asbestos workers, subcontracted workers (e.g., cleaners, carriers, security), and cohabiting relatives. Its gradual construction should prioritize regions where asbestos facilities or mines have operated and where the health systems are the strongest and more developed. Then, successful local experiences could be improved, shared, and replicated in other regions [21].

Moreover, it is essential to ensure quality control of the information related to asbestos exposure and workers’ health care and surveillance entered onto the system, including periodic monitoring, test results, ARD diagnosis, duplicate checking, etc. These definitions and procedures must be guided by the environmental and occupational health surveillance areas of the Ministry of Health. It may help to reduce the high rates of ARD deaths underreporting in Brazil, where about one-third of ARD deaths are overlooked by mortality surveillance systems [22]. It was noted that the number of deaths from mesothelioma in Brazil is lower than in other countries where asbestos use was lower, suggesting significant underreporting [23].

The system is still very dependent on Cerest’s actions for carrying out surveillance actions, secondary care, patient follow-up, and test results. Therefore, agreements between decision-makers at different levels are needed to promote and decentralize the activities of specialized health services. Promoting dialogue and cooperation between health services is challenging for an underfunded universal health coverage system such as SUS and requires continuous support and stimulation.

The definitive migration of the system to the DataSUS platform is necessary given that, once officially integrated into the SUS infrastructure, the system can be automatically financed and maintained. However, there must be management coordination between the DataSUS team and the environmental and occupational health areas of the Ministry of Health to guarantee differentiated levels of access. This coordination is also necessary to establish procedures for the system’s maintenance and improvement according to the service’s needs.

There are additional challenges related to workers’ health surveillance, such as ensuring the identification of all workers exposed and formerly exposed to asbestos, registering these workers in the system, and long-term monitoring, especially of those with higher vulnerability, those working on asbestos removal, and informal workers (e.g., workers from mines, demolition companies, the construction sector, and transportation).

Another critical challenge is establishing and strengthening a network of secondary and tertiary health services in all regions capable of providing comprehensive care for exposed individuals since identified cases of ARD may require epidemiological investigations and specialized procedures. Thus, it is necessary to define a clear healthcare flow for ARD, protocols, and responsible actors, and it must be disclosed to health services at all levels. Lastly, it is necessary to move past data entry on the system to the analysis of epidemiological information that may be useful to support decision-making and the promotion of preventive and healthcare actions.

Identifying sporadic or clusters of ARD cases could help to determine potential asbestos exposure sources and risks for workers and the general population and guide the development and implementation of risk management and communication plans. For this to happen, dedicated staff and coordination between health services involved in a patient’s diagnosis and treatment are essential. An adequate risk mitigation strategy, based on long-lasting multidisciplinary plans and periodic updates, can help guide and increase awareness of affected communities, decision-makers, victims’ organizations, and trade unions [21].

In addition to several challenges presented for the system’s implementation, Datamianto has some limitations. Some workers’ clinical and exposure data is self-reported, and may present recall and information bias, especially if a long time elapses between asbestos exposure and healthcare. Patients’ clinical information and diagnoses are filled in by trained healthcare workers, but the occurrence of biases, misinformation, and classification errors cannot be ruled out. Thus, continuous training for health workers and information quality control are essential. Moreover, the system’s operation needs human resources and technological structure (with computers and internet), which are not always available in public health services. Lastly, this information system’s use does not solve the complexity of providing comprehensive care for workers exposed to asbestos in Brazil. In this sense, the system’s information needs to be effectively used to improve asbestos-related health policies and actions.

On the other hand, there are several opportunities and possibilities in using Datamianto for facilitating the workers’ healthcare and surveillance; most benefits also apply to the non-working population.

Datamianto can help with the regular monitoring of workers exposed to asbestos, and with the efficient management of the necessary procedures within SUS. It helps to keep updated information about companies and workers and emits alerts to guide the right time for the workers’ monitoring. Moreover, it can support the monitoring of companies’ compliance with legislation, including those related to the execution of mandatory periodic appointments and worker examinations.

The system enables data storage and analysis and can provide helpful information and near real-time reports to support decision-making and promote workers’ preventive and healthcare actions. It can support early diagnosis by monitoring exposed workers and help with the epidemiological investigation of diseases and deaths possibly related to asbestos, through the sharing of registries between reference health services for investigation. This makes it possible to perform joint analysis, discuss cases, and define diagnoses, involving health services from different locations. This feature is of special importance in Brazil due to the intensive population displacement, and the lack of specialized services in some states and municipalities.

Datamianto can also promote the active surveillance of exposed workers, which should be facilitated by the organization of information on the system. It may strengthen the communication and cooperation between health services, avoiding misclassification and duplication of examinations, and facilitating mutual assistance in diagnosis and notification. Furthermore, it is possible to use the Datamianto information to conduct retrospective epidemiological analysis, thus exploring possible causal relations between a worker’s exposure to asbestos and health effects.

Thus, it may help to fill a knowledge gap related to the burden of ARD in Brazil, and to foster the implementation of the national plan for structuring the actions and health services required for comprehensive health care of Brazilian workers. Finally, the system’s evaluation by the health services was very positive and, therefore, it could be adapted or expanded to other countries and other occupational or environmental exposures.

## 4. Conclusions

Ensuring long-term comprehensive healthcare and surveillance for workers exposed to asbestos in Brazil remains an enormous challenge. In this context, the Datamianto system was recently developed to be used by the Ministry of Health for the healthcare and surveillance of workers and the general population exposed to asbestos. The system can generate short, medium, and long-term results, but its significance, applicability, and longevity will depend on the efforts aimed at its implementation and improvement.

To succeed, the worker health surveillance agenda in Brazil needs to be strengthened and overcome historical organizational and operational challenges, such as the lack of human, technological, and financial resources within SUS. Other challenges include companies’ lack of commitment to comply with the legislation, and the need to provide guidance and training for using the system and conducting worker surveillance, ensuring proper maintenance of the system and quality control of information, and establishing a network of health services for the comprehensive care of exposed workers.

Nevertheless, the system can contribute to longitudinal health monitoring and to the treatment of exposed workers, promoting dialogue and cooperation between health services, supporting law enforcement and epidemiological investigation to help decision-making, and guide the formulation of assertive strategies. Therefore, Datamianto can support and qualify the healthcare and surveillance of exposed workers and ARD in Brazil, helping to protect the workers and promote their health and quality of life.

## Figures and Tables

**Figure 1 ijerph-20-04295-f001:**
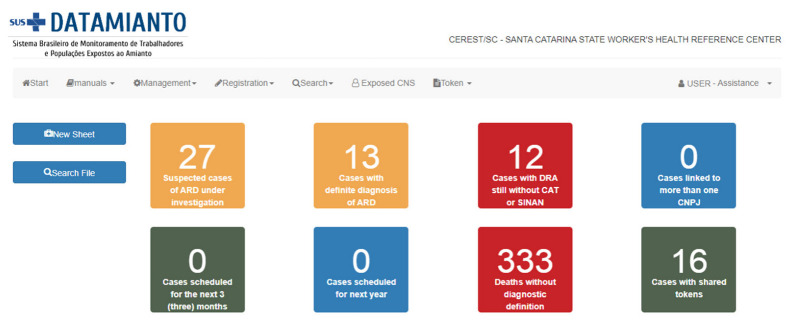
Access to the assistance module, and its alerts in the Datamianto system. Source: Datamianto system, Santa Catarina State Cerest, Brazil, 2022.

**Figure 2 ijerph-20-04295-f002:**
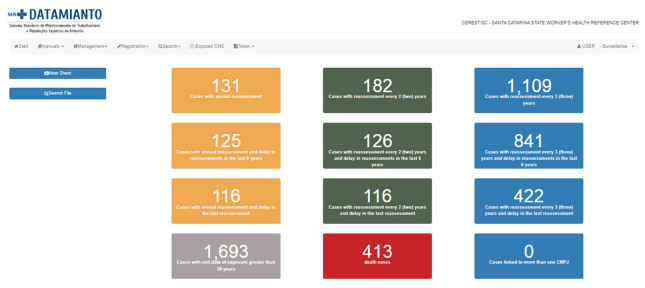
Access to the surveillance module, and its alerts in the Datamianto system. Source: Datamianto system, Santa Catarina State Cerest, Brazil, 2022.

**Figure 3 ijerph-20-04295-f003:**
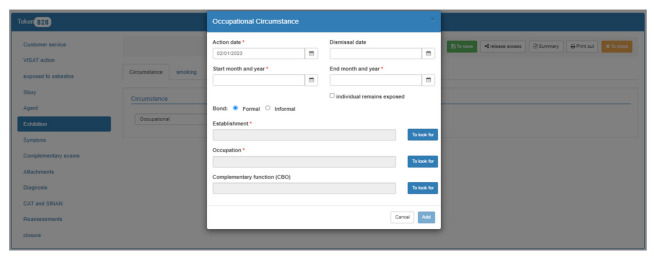
Example of worker’s information registered in the Datamianto system, 2022. Source: Datamianto system, Brazil, 2022.

**Figure 4 ijerph-20-04295-f004:**
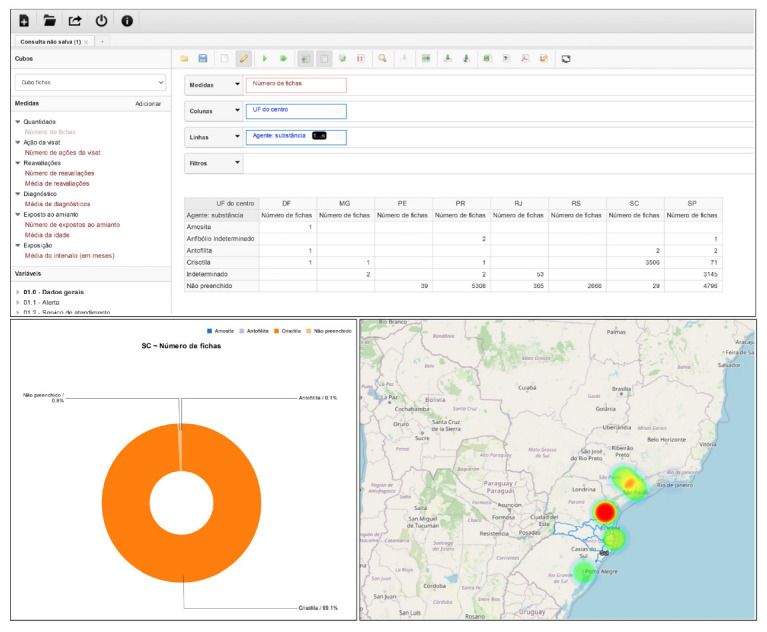
Output examples from the Datamianto system’s BI module. Notes: dynamic table of cases registered by each State, by exposure agent (**top**); chart displaying epidemiological data (**bottom left**); georeferenced heat map for registered cases by municipality/region, in which darker colors indicate more registered cases (**bottom right**). Source: Datamianto system, Brazil, 2022.

## Data Availability

Data sharing not applicable.

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
