# Peer review of "The Brazilian System for Monitoring Workers and General Population Exposed to Asbestos: Development, Challenges, and Opportunities for Workers’ Health Surveillance"

_ijerph, 2023, doi:10.3390/ijerph20054295_

Round 1
Reviewer 1 Report
The paper present a program implemented in Brazil for checking the exposure to asbestos. Despite the program looks interesting and, as the authors reported, can generate interesting epidemiological information about the exposure to asbestos, right now, it is just a program to be implemented. I think in this sense and despite the analyses of the results of this program can be within the scopes of the journal, I do not think the mere description of the program falls within this scope. It like the document but I do not think currently it is a ‘research paper’. Section like Results or Conclusions are absolutely artificial here (perhaps the name could ‘expected results’).
Author Response
Dear reviewer, thanks for your comments. To clarify, the system has already been implemented within a construction process with recurrent usability tests and data analysis validity. At this time, we do not have robust results to present, but the aim of this descriptive article is to show that it is a validated system, already in operation in some centers of care and surveillance in worker health and that it has variables that will feed reports and help analyzing epidemiological data on exposure to asbestos throughout Brazil. Lastly, it is not classified as ‘Research paper’ but as ‘Article’, and it fits perfectly to the special issue “Epidemiological Surveillance Systems of Asbestos-Related Diseases” of the IJERPH’s section: Occupational Safety and Health.

Reviewer 2 Report
It is an interesting article. I would only recommend the following:
1-Reference is made to latency periods. It is advisable to indicate the period to which it refers (e.g years) (Lines 60, 80, etc).
2-Indicate any of the workplace control measures referred to (line 62)
3-Further on it refers to 30 years (line 74) Is that the latency time? Clarify
4-it is said short time. What time do you mean? Months...(line 85)
5-The two modules that the Datamiento: Assistance and Surveillance include bibliographic justification.
6-I miss the discussion
7-Section 3.5 Main challenges and opportunities for worker’s health surveillance. I see it as practical implications, I should go after the conclusions.
Author Response
Dear reviewer, many thanks for your comments. We have considered them in this version, and understand that the manuscript’s quality improved substantially. A point-by-point reply is provided in the attached document. Best Regards.

Reviewer 3 Report
Dear Authors,
I have read your manuscript with interest. Asbestos exposure is still a great concern in the world specially in countries like Brazil where asbestos production have been so important. In general I think this descriptive manuscript is interesting for physicians and the health systems involved in the prevention and assistance of asbestos exposed subjects. My comments are mainly focused on details that can hep readers to understand how this database really works.
- In the introduction, I suggest to include some words indicating the relevance of these kind of monitoring systems for the health services around the world.
- I would shorten the first paragraph of the introduction.
- Methods: Who is responsible for the data introduction? In some registries, there are variables requiring some expertise to discriminate the criteria to be taken into account. If the participation of doctors is not needed, what is the risk of making mistakes in the database?
In the same line, how much time is needed to compliment a patient's registry?
In page 8 it is said that Datamianto has been used in 28 health services in Brazil. Does it mean that 28 states have used it? Please clarify the meaning of "Health services"
- A model of the database should be displayed in a supplementary material
- Page 9: I undertand that this database is offered to doctors ans companies, but it should be stated that it is a voluntary option if this is the case.
- In page 9, when exposing the number of exposed workers already included, it would be interesting to know the total number of exposed workers to understand the degree of implementation of the database.
- In discussion, I think a paragraph of limitations of this system should be included.
Author Response

(The authors gave the same response as above.)

Round 2
Reviewer 1 Report
No comments.